# Local Recurrence of Premalignant and Early Malignant Rectal Polyps Treated by TEM—A Single-Center Experience

**DOI:** 10.3390/jcm14010080

**Published:** 2024-12-27

**Authors:** Muhammad Khalifa, Rachel Gingold-Belfer, Nidal Issa

**Affiliations:** 1Department of Surgery, Rabin Medical Center-Hasharon Hospital, Faculty of Medicine, Tel Aviv University, Petach Tikva 49100, Israel; nidalissa@clalit.org.il; 2Department Gastroenterology, Rabin Medical Center-Hasharon Hospital, Faculty of Medicine, Tel Aviv University, Petach Tikva 49100, Israel; rachelgingoldbelfer@gmail.com

**Keywords:** transanal endoscopic microsurgery, early rectal cancer, high grade dysplasia, local recurrence

## Abstract

**Background:** Transanal endoscopic microsurgery (TEM) is a minimally invasive approach for excising rectal polyps, particularly those with high-grade dysplasia (HGD) or early-stage rectal cancer (T1). This study aimed to evaluate the recurrence risk and its associated factors in patients treated with TEM for HGD and T1 rectal tumors. **Methods:** A retrospective review was conducted on 79 patients who underwent TEM for rectal lesions at Rabin Medical Center-Hasharon Hospital from 2005 to 2019. Data collected included demographics, tumor characteristics, and follow-up outcomes, with specific focus on tumor size, resection margins, mucin production, and distance from anal verge (AV). Separate and unified analyses were performed to assess the recurrence risk factors for both HGD and T1 patients. **Results:** Sixty-three patients were included in the final analysis. In the unified analysis, larger tumor size was significantly associated with increased recurrence risk (OR = 2.27, *p* = 0.028), and mucin production was a strong predictor of recurrence in the T1 group and combined analysis (*p* = 0.0012 and *p* = 0.014, respectively). Distance from AV demonstrated a borderline association with recurrence (*p* = 0.053). **Conclusions:** Larger tumor size and mucin production are significant predictors of recurrence in TEM-treated rectal polyps. Personalized follow-up and postoperative management are essential for patients with these risk factors to reduce the recurrence risk.

## 1. Introduction

Rectal polyps, encompassing a spectrum from benign to premalignant and malignant lesions, represent a critical focus in colorectal cancer prevention and management. High-grade dysplasia (HGD) and early-stage rectal cancer (T1) are two entities along this spectrum that carry distinct risks of progression and recurrence [1]. Recent advances in minimally invasive surgical techniques, such as transanal endoscopic microsurgery (TEM), have revolutionized the management of rectal polyps. TEM enables the precise local excision of rectal lesions with a high degree of accuracy and minimal impact on surrounding structures, which is particularly valuable for patients with HGD or early-stage rectal cancers who may benefit from organ-preserving approaches [2,3]. Nonetheless, the risk of local recurrence remains a significant challenge, especially for specific subgroups based on histopathological characteristics [4].

The recurrence of rectal lesions after TEM is influenced by several factors, with tumor size, resection margin status, and histological features emerging as critical determinants [5,6]. Larger tumors have been associated with higher recurrence rates due to the technical challenges in achieving complete local excision and the possible presence of residual cancer cells beyond the visualized [7,8]. Moreover, involved or narrow resection margins have been highlighted in various studies as a predictor of recurrence, given the likelihood of residual microscopic disease that may not be completely removed by TEM [9,10,11]. The presence of mucin production in rectal tumors has also garnered attention due to its potential association with a more aggressive tumor phenotype and increased recurrence risk, though findings have been inconsistent and vary across different studies [12].

Differentiation between HGD and T1 cancers introduces further complexity in assessing the recurrence risk. While both groups share certain pathological characteristics, T1 cancers exhibit invasive behavior, necessitating a different approach to postoperative surveillance and risk assessment [13]. The current study aimed to bridge these gaps by analyzing factors associated with recurrence separately in the HGD and T1 groups and conducting a unified analysis that integrated both groups. By evaluating the tumor characteristics, this study sought to elucidate the risk factors for recurrence after TEM, contributing valuable insights to guide clinical decision-making and postoperative surveillance.

## 2. Methods

Approval for this retrospective study was obtained from the Institutional Review Board of the Rabin Medical Center, with a waiver of informed consent. A comprehensive review of the EMRs was conducted for all patients who underwent transanal endoscopic microsurgery (TEM) at Hasharon Hospital, Rabin Medical Center, between January 2005 and December 2019. Patients who underwent rectal lesion excision or resection through any approach other than TEM were excluded from the study. Additionally, patients with a history of prior colon resection for malignancy or previous chemotherapy were excluded to ensure that tumor recurrence was not influenced by other treatments.

The indications for TEM included rectal polyps not amenable to endoscopic resection, incomplete endoscopic resection, and recurrence after endoscopic resection. Lesions selected for TEM were clinically and radiologically diagnosed as benign adenomas, premalignant adenomas, or early T1 rectal cancer.

Following an initial analysis of all eligible patients, a further subset analysis was performed, focusing exclusively on cases with high-grade dysplasia (HGD) or early-stage adenocarcinoma (T1). The risk of recurrence was evaluated by analyzing the relationship between tumor recurrence and specific tumor-related factors including tumor size, surgical margin status, mucin production, and tumor distance from the anal verge (AV).

All TEM procedures were performed by a single surgeon using original Richard Wolf (Knittlingen, Germany) TEM equipment under general anesthesia. The details of the technique have been previously described elsewhere [14].

A thorough review of each patient’s full medical record was conducted to collect data on the demographics, tumor characteristics, tumor location, surgical indications, histopathological findings, postoperative outcomes, complications, and recurrence.

## 3. Statistical Analysis

Data were analyzed with the Statistical Package for the Social Sciences (SPSS) version 22.0.

Descriptive statistics: Summary statistics were calculated for each group including the means and standard deviations for the continuous variables and frequencies for the categorical variables. Univariate analysis was used for comparison, and Chi-square tests were conducted to assess the associations between recurrence and categorical predictors (tumor size binary, tumor margins binary, mucin production). The *t*-tests were conducted to compare continuous predictors (tumor size, tumor margins, and distance from AV) between the recurrence and no recurrence groups. The latter variables were tested in a multivariable logistic regression to assess the association between specific pathological characteristics of the group and local recurrence. Odds ratios (ORs) were calculated to quantify the effect size of each predictor on recurrence. A *p*-value < 0.05 was considered significant.

## 4. Results

### 4.1. Baseline Characteristics

Between January 2005 and December 2019, a total of 138 patients underwent transanal endoscopic microsurgery (TEM) for rectal lesions initially diagnosed preoperatively as either adenoma with dysplasia or early-stage adenocarcinoma. Among these, 55 patients were excluded based on a final pathology of low-grade dysplasia (LGD), and an additional four patients were excluded due to prior chemotherapy treatment. The remaining 79 patients, forming the study cohort, were diagnosed with adenoma with high-grade dysplasia (HGD) or adenocarcinoma tumors, out of which 63 patients were included in the final analysis. The flowchart of patient selection is shown in Figure 1.

The mean age of the patients was 69 years (±11), with a male representation of 48% (38 patients). In terms of preoperative physical status, 67% of the patients were classified as American Society of Anesthesiologists (ASA) score 1–2, while 33% were classified as ASA score 3–4. Tumor characteristics demonstrated an average diameter of 2.7 cm (±1.5 cm) with a range from 0.6 to 8 cm. The mean distance from the anal verge (AV) was 9.3 cm (±3.2 cm), spanning from 3 to 18 cm. Tumor location was distributed as follows: 19% in the anterior position, 25% posterior, 27% on the right side, 24% on the left side, and 5% non-reported (NR). Preoperative imaging was employed variably, with endorectal ultrasound (ERUS) utilized in 86% of cases, computed tomography (CT) in 60%, and magnetic resonance imaging (MRI) in 40%. The mean follow-up duration was 46 months (±43 months). Table 1 summarizes the demographic and preoperative tumor characteristics of the patients.

### 4.2. Tumor Characteristics on Final Pathology

The final pathology analysis of the tumors revealed the following characteristics. Thirty-two (40%) tumors were classified as adenomas with high-grade dysplasia (HGD), thirty-one (39%) as T1 adenocarcinomas, fourteen (18%) as T2 adenocarcinomas, and two (3%) as T3 adenocarcinomas. Mucinous features were present in nine (11%) tumors. All locally advanced tumors (T2, T3) were misdiagnosed preoperatively as clinically T1 lesions according to Endo-Rectal US imaging.

The mean tumor size was 2.7 cm (±1.5 cm) with a range of 0.6 to 8 cm. Surgical margins averaged 4.9 mm (±3.6 mm). Notably, clear margins of ≥3 mm were achieved in 72% of cases, while 22% had margins <3 mm, and 6% had involved margins. Neither lympho-vascular invasion nor peri-neural invasion was detected in the specimens. Table 2 summarizes the tumor characteristics according to the final pathology.

### 4.3. Postoperative Management

Postoperative management strategies were tailored based on the final pathological stage of the tumor. Patients diagnosed with high-grade dysplasia (HGD) or T1 stage tumors were managed conservatively with follow-up alone, as no additional surgical or radiotherapy interventions were deemed necessary.

For T2 stage tumors, five patients received follow-up only due to their refusal of radical surgery or adjuvant therapy, eight patients underwent further intervention with total mesorectal excision (TME), and one patient received radiotherapy. Among the five patients who declined surgery or adjuvant treatment, three experienced tumor recurrence and subsequently died from metastatic disease; notably, two of these patients had positive margins on final pathology. For T3 stage tumors, two patients proceeded to TME surgery. Table 3 summarizes the postoperative management of all patients.

### 4.4. Analysis of the HGD and T1 Groups

Table 4 presents the baseline characteristics of patients within the HGD and T1 groups. The mean tumor size was larger in the T1 group compared to the HGD group, though a full statistical comparison between groups was not the focus of this study. Tumor margins also showed some variability between groups, with the T1 group having a slightly larger mean margin. Distance from the anal verge (AV) was similarly distributed between groups. Mucin production was more prevalent in the T1 group, whereas differentiation was only applicable for the T1 group, showing a higher percentage of well-differentiated tumors.

Tumor recurrence was identified in four patients (12.5%) within the HGD group. Among these, one patient experienced recurrence with high-grade dysplasia (HGD) and underwent successful re-treatment with transanal endoscopic microsurgery (re-TEM). The remaining three patients developed T1 invasive carcinoma and were subsequently managed with total mesorectal excision (TME). In the T1 group, two patients experienced recurrence as locally advanced T3 invasive carcinoma, both of whom were treated with TME.

### 4.5. Univariate Analysis for Recurrence

The association between tumor recurrence risk and the specified tumor characteristics was examined. In the univariate analysis (Table 5), no statistically significant relationship was identified between any of the four tumor factors and recurrence risk within the HGD patient group. However, mucin production demonstrated a significant association with tumor recurrence in both the T1 group (*p* = 0.0012) and the combined patient analysis (*p* = 0.014). Additionally, distance from the anal verge (AV) exhibited a borderline significant association with recurrence in the combined analysis (*p* = 0.053).

### 4.6. Multivariate Analysis for Recurrence

Table 6 provides the results of the logistic regression analyses estimating the odds of tumor recurrence based on the tumor characteristics. Key findings include:Tumor size: In the combined analysis, tumor size showed a statistically significant association with recurrence (OR = 2.27, 95% CI: 1.09–4.73, *p* = 0.028). This indicates that each unit increase in tumor size approximately doubles the odds of recurrence. The HGD group showed an OR of 5.95 (95% CI: 0.77–46.11, *p* = 0.088), suggesting a potential, though non-significant, risk of recurrence with increasing tumor size.Tumor margins: Across all analyses, the tumor margins did not show a statistically significant association with recurrence. In the combined analysis, the OR was 1.12 (95% CI: 0.79–1.57, *p* = 0.526), indicating a minimal effect on recurrence risk.Distance from AV: Distance from AV demonstrated a borderline significant association with recurrence in the combined analysis (OR = 0.65, 95% CI: 0.42–1.00, *p* = 0.052). This finding suggests that tumors located further from the AV might have a lower risk of recurrence.
jcm-14-00080-t006_Table 6Table 6Multi-variate analysis results for risk of tumor recurrence.VariableHGD Odds Ratio (95% CI) *p* ValueN = 32T1 Odds Ratio (95% CI) *p* ValueN = 31Combined Odds Ratio (95% CI) *p* Value N = 63Tumor size5.95 (0.77–46.11)0.0881.10 (0.24–4.98)0.9022.27 (1.09–4.73)0.028Tumor margins2.74 (0.63–11.86)0.1770.86 (0.50–1.49)0.5991.12 (0.79–1.57)0.526Distance from AV0.26 (0.03–2.33)0.2280.69 (0.34–1.37)0.2890.65 (0.42–1.00)0.052


## 5. Discussion

The findings from this study provide important insights into the recurrence risk associated with TEM for rectal polyps with HGD and T1 cancer. In our analysis, tumor size emerged as a significant predictor of recurrence in the unified model, suggesting that larger tumors may require closer postoperative surveillance regardless of the histological classification as HGD or T1 [15]. This result aligns with previous studies that have demonstrated an association between tumor size and recurrence risk in colorectal polyps, as larger tumors pose technical challenges for complete excision and often harbor more aggressive or advanced histological features [16]. From a clinical perspective, this finding emphasizes the need for comprehensive preoperative assessment and potentially more aggressive postoperative follow-up for patients with larger tumors undergoing TEM.

The role of distance from the AV as a predictor of recurrence, which was found to be marginally significant in the unified model, is also notable. Although the association did not reach conventional levels of significance, the negative coefficient indicates that greater distance from the AV might reduce the recurrence risk. This observation could be attributed to the technical nuances of performing TEM in proximity to the anal sphincter, where achieving clear resection margins can be more challenging due to anatomical constraints. This trend supports the growing body of evidence suggesting that lesions closer to the AV may require enhanced follow-up protocols and the consideration of additional therapeutic modalities if complete resection is uncertain [17].

Mucin production was found to be significantly associated with recurrence, which could have implications for histopathological assessment in determining the recurrence risk. Mucinous features have been associated with a more aggressive tumor biology in various malignancies including colorectal cancer due to mucin’s ability to create a microenvironment conducive to tumor growth and spread [12,18]. In our study, the presence of mucin production in rectal polyps significantly increased the recurrence risk, suggesting that mucin production may serve as a valuable marker for identifying high-risk lesions. This finding supports the need for a meticulous pathological evaluation of mucinous features in rectal polyps, and may justify more intensive postoperative surveillance for patients with mucin-producing lesions.

The lack of significant association between the tumor margins and recurrence in both groups and the unified analysis warrants further discussion. While positive or narrow margins are widely considered as a risk factor for recurrence in colorectal surgery [19], our study did not find a significant effect, which may be due to the relatively small sample size or variability in pathological assessment across cases. Nevertheless, it is important to consider that in real-world practice, margin status is still a critical factor in determining the adequacy of resection and guiding further treatment. Other studies with larger sample sizes and a more standardized approach to margin assessment may be necessary to conclusively determine the role of margin status in recurrence risk after TEM.

In terms of group-specific findings, the HGD group showed a trend toward higher recurrence risk associated with tumor size, though this was not statistically significant. This could suggest that tumor size is a relevant factor even in premalignant lesions, supporting the notion that larger polyps, even those classified as HGD, may carry a greater risk of recurrence due to residual tissue or potential malignant transformation [5]. The T1 group, in contrast, demonstrated no significant association between tumor recurrence and any of the analyzed individual predictors, except for mucin production. Mucin production showed a strong and statistically significant association with recurrence, suggesting its critical role in tumor biology and progression. This lack of significance may be due to the invasive nature of T1 lesions, where other molecular or genetic factors not assessed in this study could play a more critical role in recurrence.

It is worth mentioning that in the T1 group, no cases displayed adverse prognostic features such as lympho-vascular invasion or perineural invasion, which are typically associated with poorer outcomes in rectal cancer. The absence of these high-risk features likely contributed to the favorable outcomes observed in this cohort and may limit the generalizability of these findings to patients with more aggressive T1 lesions. Additionally, the depth of tumor invasion within the submucosa, as categorized by the Kikuchi classification [20], was not emphasized in this study due to a lack of available data. Given that deeper submucosal invasion (Sm3) is associated with increased recurrence risk, this missing information may have impacted the study results by omitting an important prognostic factor. Future studies should incorporate SM classification data to provide a more comprehensive assessment of recurrence risk in early-stage rectal cancers treated with TEM.

The study also highlights the importance of individualized postoperative management strategies based on tumor pathology. Patients with HGD and T1 tumors were successfully managed with TEM alone, with no additional interventions required. In contrast, patients with T2 stage tumors presented a more complex management profile, as follow-up alone was insufficient for many cases. Of the five patients with T2 tumors who declined additional treatment, three developed metastatic recurrence and ultimately succumbed to the disease, with two of these patients exhibiting positive margins on final pathology. This outcome emphasizes the risks associated with declining recommended radical surgery or adjuvant therapy in cases with higher pathological stages or positive margins [21]. Patients with T2 and T3 tumors should be counseled regarding the increased recurrence risk if additional treatments are not pursued, particularly when pathology reveals positive or close margins.

Regarding the recurrence rates following TEM, our study demonstrated a local recurrence rate of 12.5% for HGD and 6% for early favorable T1 rectal cancer. These results are in line with previously published data, where the recurrence risk for adenomas ranges from 2.4% to 16%, while local recurrence rates following the local excision of adenocarcinomas have been reported to range between 3.8% and 22% [4,22].

It is worth noting that alternatives to transanal endoscopic microsurgery (TEM), such as transanal minimally invasive surgery (TAMIS) and robotic transanal surgery, have demonstrated comparable surgical and oncological outcomes in the management of rectal polyps and early-stage rectal cancers. TAMIS, while more cost-effective and accessible due to its reliance on standard laparoscopic instruments, may lack the precision and visualization provided by TEM’s dedicated optical system [23]. Robotic transanal surgery offers enhanced dexterity and superior 3D imaging, making it particularly beneficial for complex resections; however, its high cost and limited availability remain significant drawbacks [24]. Although evaluating these alternatives was beyond the scope of this study, TEM continues to offer distinct advantages including unparalleled precision, a long-established safety profile, and reliable outcomes, particularly in anatomically challenging cases.

## 6. Limitations

This study has several limitations that should be acknowledged. First, the relatively small sample size may limit the applicability of the findings and reduce the statistical power to detect significant associations between tumor-related factors and recurrence. Second, the absence of detailed information on submucosal invasion depth and extent, particularly using the Kikuchi classification (Sm1, Sm2, Sm3), may have limited the ability to fully evaluate recurrence risk in the T1 group, as submucosal invasion is a critical prognostic factor in early rectal cancer. Third, inconsistencies in histopathological evaluation including variability in margin measurement and the assessment of mucin production could introduce bias and affect the accuracy of the findings. Additionally, as a retrospective study, it is subject to inherent limitations, such as potential missing data or incomplete records, which may have influenced the results. Finally, the analysis focused on traditional clinical and pathological factors without incorporating molecular or genetic markers, which are increasingly recognized as important determinants of recurrence risk.

## 7. Clinical Implications and Future Directions

These findings underscore the importance of a tailored approach to managing rectal polyps with HGD and T1 cancers treated by TEM [25]. The significant association of tumor size with recurrence suggests that tumor dimensions should be a key factor in preoperative planning and postoperative follow-up. Patients with larger tumors, regardless of classification as HGD or T1, may benefit from enhanced imaging, endoscopic surveillance, or even the consideration of adjunctive treatments to reduce the recurrence risk. Moreover, the association between mucin production and recurrence highlights the value of detailed pathological assessment in identifying high-risk cases that warrant closer monitoring.

Future research should aim to include larger cohorts and prospective designs to validate these associations and provide more robust risk stratification tools. Additionally, exploring molecular markers, such as genetic mutations or biomarkers specific to rectal tumorigenesis, could further refine recurrence risk prediction and help personalize postoperative care for patients undergoing TEM.

## Figures and Tables

**Figure 1 jcm-14-00080-f001:**
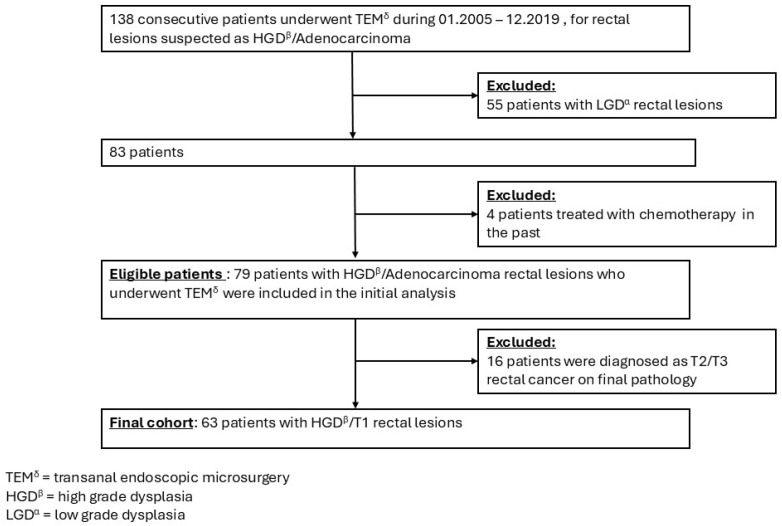
The flowchart of patient’s selection.

**Table 1 jcm-14-00080-t001:** Patient demographics and preoperative characteristics.

N = 79	
Age (years)	69 ± 11
Male (%)	38 (48)
ASA score (%)	
1–2	53 (67)
3–4	25 (33)
Preoperative tumor characteristics	
Diameter (cm), (range)	2.7 ± 1.5 (0.6–8)
Distance from AV (cm), (range)	9.3 ± 3.2 (3–18)
Location (%)	
Anterior	15 (19)
Posterior	20 (25)
Right	21 (27)
Left	19 (24)
NR	4 (5)
Preoperative imaging (%)	
ERUS	68 (86)
CT	48 (60)
MRI	32 (40)
Follow-up, mean, months	46 ± 43

**Table 2 jcm-14-00080-t002:** Tumor characteristics based on the final pathology report.

N = 79	
HGD (%)	32 (40)
T1 (%)	31 (39)
T2 (%)	14 (18)
T3 (%)	2 (3)
Mucinous (%)	9 (11)
Size (cm), (range)	2.7 ± 1.5 (0.6–8)
Margins (mm)	4.9 ± 3.6
Involved (%)	5 (6)

**Table 3 jcm-14-00080-t003:** Postoperative management of all patients.

	Follow-Up	TME	Radiotherapy
HGD (%)	32 (100)	-	-
T1 (%)	31 (100)	-	-
T2 (%)	5 (36)	8 (57)	1 (7)
T3 (%)	-	2 (100)	-
Overall	68	10	1

**Table 4 jcm-14-00080-t004:** Tumor characteristics and recurrence rate classified among the HGD and T1 groups.

Variable	HGD Group (Mean ± SD) N = 32	T1 Group (Mean ± SD) N = 31	*p*-Value
Tumor size (cm)	2.71 ± 1.74	2.39 ± 1.06	0.392
Tumor margins (mm)	3.58 ± 2.46	6.23 ± 4.14	0.003
Distance from AV (cm)	9.06 ± 3.60	9.48 ± 3.03	0.621
Mucin production, n (%)	0	3 (9.7)	0.076
Recurrence, n (%)	4 (12.5)	2 (6)	0.41

**Table 5 jcm-14-00080-t005:** Univariate analysis for tumor recurrence vs. tumor characteristics.

Variable	HGD *p*-Value N = 32	T1 *p*-Value N = 31	Combined *p*-Value N = 63
Mucin production ^†^	1.000	0.0012	0.014
Tumor size ^‡^	0.316	0.712	0.218
Tumor margins ^‡^	0.900	0.256	0.216
Distance from AV ^‡^	0.091	0.528	0.053

^†^ Chi-square test. ^‡^
*t*-test.

## Data Availability

The data that support the findings of this study are available from the corresponding author, M.K., upon reasonable request.

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
