# Peer review of "Local Recurrence of Premalignant and Early Malignant Rectal Polyps Treated by TEM—A Single-Center Experience"

_jcm, 2024, doi:10.3390/jcm14010080_

Round 1

Reviewer 1 Report

Comments and Suggestions for Authors

The authors report on the effectiveness of transanal endoscopic microsurgery (TEM) as a means to treat and control early-stage colorectal cancer (CRC). Authors summarize the retrospective analysis of patients that underwent TEM at a single center between 2005-2019 to determine the effectiveness of TEM as a means to control early-stage CRC. While the manuscript indicates that TEM is an effective method to control CRC, the authors do not indicate or suggest how impactful TEM is when compared to current or alternative strategies. These comparisons would be beneficial to incorporate to improve signficance and interest.

Major questions:

Authors stratified their patient cohort between HGD and T1 CRC, and show difference between cohorts in recurrence, but do not reveal how many patients from each group recurred. It would also be of interest to reveal whether patients that recurred showed signs of advancement in stage.

It would be interesting to compare TEM patient recurrence with patients that undergo alternative strategies of treatment. Do you see reduction in local recurrence or is there minimal difference, but TEM is less invasive for patients?

Minor questions:

Total patients was 133, excluding 59. However, patients in analysis was 79, indicating additional patients were included in the cohort.

ASA (line 95) is not defined.

Author Response

Comments 1:  how many patients from each group recurre? It would also be of interest to reveal whether patients that recurred showed signs of advancement in stage.

Response 1: Thank you for pointing this out. We added the missing information regarding recurrence rate and the stage of recurrence in the results section (lines 143-149).

Comments 2: It would be interesting to compare TEM patient recurrence with patients that undergo alternative strategies of treatment.

Response 2: Transanal endoscopic microsurgery (TEM) is a highly effective technique for local excision of rectal lesions. Local excision, in general, represents a minimally invasive approach that has been demonstrated to be both safe and effective for managing benign and early-stage rectal cancers, offering the advantage of avoiding the need for radical surgery in selected patients. Alternative approaches to TEM, including transanal minimally invasive surgery (TAMIS) and robotic-assisted TEM, have emerged in recent years. A detailed comparison of these techniques, highlighting their respective advantages and limitations, is provided in the discussion section (lines 252–262).

Comments 3: Total patients was 133, excluding 59. However, patients in analysis was 79, indicating additional patients were included in the cohort.Response 3:

Response 3: we fix this mistake.

Comment 4: ASA (line 95) is not defined.

Response 4: we fix it.

Reviewer 2 Report

Comments and Suggestions for Authors

The authors present in a overall well structured and comprehensive manner the possible correlations between tumor characteristics and the recurrence risk for HGD and T1 patients who underwent TEM in their center. The study is a single center experience, with a sufficient population size and mean follow up. Statistical methods are good, but improvable. Overall presentation can also be improved. The conclusions are supported by the results. In order to increase the strength of the manuscript, I suggest the following:

1) the authors should specify their indications for TEM treatment in the methods section

2) the authors should include the mentioned figure 1, as it is missing in the uploaded manuscript

3) the authors should specify the preoperative staging of the patients who underwent TEM and had a pT of T2-3 in the results section

4) table 5 should be revisited, as the analysis presented is a descriptive analysis (comparison of means) and not a univariate analysis (regression)

5) table 6 should report the p-values for the calculated odds ratio

6) tables should be reformatted to respect the MDPI format and statistically significant p-values should be in bold

7) the authors should increase the number of references

Author Response

Comments 1: the authors should specify their indications for TEM treatment in the methods section.

Response 1: Thank you for pointing this out. we added the indicatins for TEM in the Methods Section (lines 60-63)  

Comments 2: the authors should include the mentioned figure 1, as it is missing in the uploaded manuscript.

Response 2: we added figure 1. To the manuscript.

Comments 3:  the authors should specify the preoperative staging of the patients who underwent TEM and had a pT of T2-3 in the results section.

Response 3: All the patients who were diagnosed post operatively as T2, T3 tumors, were previously misdiagnosed as T1 by Endo-rectal US. Find the addition in the Results (lines 118-119)

Comment 4: table 5 should be revisited, as the analysis presented is a descriptive analysis (comparison of means) and not a univariate analysis (regression)

Response 4: in table 4, we did a comparison of means only for the mucin production variable. The other continuous variables were calculated by t-test. We added a clarification on the table index.

Comment 5: table 6 should report the p-values for the calculated odds ratio.

Response 5: we fix it.

Comment 6: tables should be reformatted to respect the MDPI format and statistically significant p-values should be in bold.

Response 6: we fix it.

Comment 7: the authors should increase the number of references.

Response 7: we fix it.

Reviewer 3 Report

Comments and Suggestions for Authors

This single center experience focusing on patients undergoing transanal endoscopic microsurgery for the resection of high grade dysplasia or T1 lesions aims to identify from amongst 4 tumor characteristics, namely tumor size, resection margins, tumor height, and mucin production, the feature(s) that are most predictive of lesion recurrence.

The authors do a great job reporting their selection criteria, summarizing their data, providing the readers with tables that include both the total cohort (n=79) and the subgroup that was analyzed (n=63), and the surveillance/management strategies that were implemented based on lesion's final pathology. However, the following may need to be addressed:

(1) References: The total number of references (n=17) may be too small. In addition, the most recent reference cited (number 5) dates back in 2022. The manuscript could benefit from the identification, incorporation, and discussion of more recent and a larger number of references. 

(2) Methodology: Line 76 states: "Univariate analysis was used for comparison..." and lines 80-82 state: "Significant variables were tested in a multivariable logistic regression to assess the association between specific pathological characteristics of the group and local recurrence". However, it seems that all the variables were included in the multivariable model, not only the significant ones (for example the univariate analyses in the HGD group did not identify any significant associations however a multivariate model incorporating all four variables was constructed). It may be helpful for the statement in lines 80-82 to be clarified.

(3) Results: The results are well described however, it may be useful if subsections were created. Consider creating: (1) Baseline characteristics; (2) Postoperative strategies; (4) Univariate analyses outcomes; (5) Multivariate logistic regression. The nomenclature provided is just an example, however the manuscript could benefit from breakdown of this nature.

(4) Discussion: Lines 205-207 which discuss the group-specific findings state: "The T1 group, in contrast, showed no significant association between recurrence and any of the individual predictors in our analysis." however, lines 148-149 which discuss the results of the univariate analyses for the T1 group state: "...mucin production demonstrated a significant association with tumor recurrence in both the T1 groups and the combined patient analysis." Similarly lines lines 155-157 that discuss the multivariate regression results for this subgroup state: "In the T1 group, mucin production emerged as a significant predictor, showing a strong association with increased recurrence risk (p=0.0012)." It may be beneficial for this discrepancy to be clarified.

(5) Limitations: A limitations section is missing in the "Discussion". I would highly encourage the incorporation of such a paragraph/section.

Author Response

Comments 1:  The total number of references (n=17) may be too small. In addition, the most recent reference cited (number 5) dates back in 2022. The manuscript could benefit from the identification, incorporation, and discussion of more recent and a larger number of references.

Response 1: Thank you for pointing this out. We added some more references. Unfortunately we could not find a more recent and relative references to cite.  

Comments 2  Methodology: Line 76 states: "Univariate analysis was used for comparison..." and lines 80-82 state: "Significant variables were tested in a multivariable logistic regression to assess the association between specific pathological characteristics of the group and local recurrence". However, it seems that all the variables were included in the multivariable model, not only the significant ones (for example the univariate analyses in the HGD group did not identify any significant associations however a multivariate model incorporating all four variables was constructed). It may be helpful for the statement in lines 80-82 to be clarified.

Response 2: Indeed you are right. All the variables were included in the multivariate analysis, and we fix the mistake.

Comments 3: Results: The results are well described however, it may be useful if subsections were created. Consider creating: (1) Baseline characteristics; (2) Postoperative strategies; (4) Univariate analyses outcomes; (5) Multivariate logistic regression. The nomenclature provided is just an example, however the manuscript could benefit from breakdown of this nature.

Response 3: To enhance clarity and accessibility, an explanatory paragraph has been added to elaborate on the multivariate analysis results presented in Table 6 (lines 163–179). Additionally, data on recurrence rates within each group have been incorporated to provide a more comprehensive understanding of the study outcomes (lines 143–149)

Comment 4:  Discussion: Lines 205-207 which discuss the group-specific findings state: "The T1 group, in contrast, showed no significant association between recurrence and any of the individual predictors in our analysis." however, lines 148-149 which discuss the results of the univariate analyses for the T1 group state: "...mucin production demonstrated a significant association with tumor recurrence in both the T1 groups and the combined patient analysis." Similarly lines lines 155-157 that discuss the multivariate regression results for this subgroup state: "In the T1 group, mucin production emerged as a significant predictor, showing a strong association with increased recurrence risk (p=0.0012)." It may be beneficial for this discrepancy to be clarified.

Response 4: A statistically significant association between mucin production and the risk of recurrence in the T1 group was identified in our analysis. This finding was inadvertently misrepresented in the initial discussion section. The discrepancy has been addressed and clarified appropriately (lines 223–227).

Comment 5: Limitations: A limitations section is missing in the "Discussion". I would highly encourage the incorporation of such a paragraph/section.

Response 5: thank you for pointing this out. we added a Limitaions section.

Round 2

Reviewer 1 Report

Comments and Suggestions for Authors

Authors addressed reviewer's concerns in response. Authors now reveal the rates of recurrence in the patient cohort, and discuss alternative strategies on surgical treatment of CRC patients. However, some minor concerns remain:

1) With the rate of recurrence using TEM, can authors compare rate of recurrence from their cohort with publicly available data to discuss implications of whether TEM, TME, TAMIS, or surgical resection have different rates of recurrence? If the authors are advocating the use of TEM as a primary means to treat and control early-stage CRC with minimal risk and drawbacks, the TEM recurrence rate should be compared with alternative means of treatment in addition to the discussion of pros and cons. 

2) Authors reveal that 79 patient cohort was further reduced by mis-staging of T2/T3 patients. Final n of cohort should be emphasized in results and Figure 1 flowchart.

Author Response

comment 1: With the rate of recurrence using TEM, can authors compare rate of recurrence from their cohort with publicly available data to discuss implications of whether TEM, TME, TAMIS, or surgical resection have different rates of recurrence? If the authors are advocating the use of TEM as a primary means to treat and control early-stage CRC with minimal risk and drawbacks, the TEM recurrence rate should be compared with alternative means of treatment in addition to the discussion of pros and cons. 

response1: It is well established in the literature that TEM for early rectal cancer provides oncological outcomes comparable to radical TME, particularly in terms of local and distant recurrence rates. While this was not the primary focus of our study, we have included a paragraph comparing our observed recurrence rates with previously published reports on rectal cancers treated with local excision to provide additional context (lines 255-259)

comment 2: Authors reveal that 79 patient cohort was further reduced by mis-staging of T2/T3 patients. Final n of cohort should be emphasized in results and Figure 1 flowchart.

response 2: thank you for pointing this out. we changed the flowchart and added the relevant change in the results (line 96)

Reviewer 2 Report

Comments and Suggestions for Authors

The authors have addressed all my concerns. I consider the manuscript acceptable for publication.

Author Response

Thank you so much for your valuable notes.